# Adaptive recurrent vision performs zero-shot computation scaling to unseen difficulty levels

**Vijay Veerabadran**[1*]   **Srinivas Ravishankar**[1*]
**Yuan Tang**[1†]   **Ritik Raina**[1†]   **Virginia R. de Sa**[1,2]
[1] Department of Cognitive Science, [2] Halıcıoğlu Data Science Institute,
University of California, San Diego, La Jolla, CA 92093
`vveeraba@ucsd.edu`

## Abstract

Humans solving algorithmic (or) reasoning problems typically exhibit solution times that grow as a function of problem difficulty. Adaptive recurrent neural networks have been shown to exhibit this property for various language-processing tasks. However, little work has been performed to assess whether such adaptive computation can also enable vision models to extrapolate solutions beyond their training distribution's difficulty level, with prior work focusing on very simple tasks. In this study, we investigate a critical functional role of such adaptive processing using recurrent neural networks: to dynamically scale computational resources conditional on input requirements that allow for zero-shot generalization to novel difficulty levels not seen during training using two challenging visual reasoning tasks: PathFinder and Mazes. We combine convolutional recurrent neural networks (ConvRNNs) with a learnable halting mechanism based on (Graves, 2016). We explore various implementations of such adaptive ConvRNNs (AdRNNs) ranging from tying weights across layers to more sophisticated biologically inspired recurrent networks that possess lateral connections and gating. We show that 1) AdRNNs learn to dynamically halt processing early (or late) to solve easier (or harder) problems, 2) these RNNs zero-shot generalize to more difficult problem settings not shown during training by dynamically increasing the number of recurrent iterations at test time. Our study provides modeling evidence supporting the hypothesis that recurrent processing enables the functional advantage of adaptively allocating compute resources conditional on input requirements and hence allowing generalization to harder difficulty levels of a visual reasoning problem without training.

## 1   Introduction

Recurrent Neural Networks (RNNs) have emerged as a powerful tool for solving machine reasoning tasks, demonstrating impressive performance across a variety of tasks that require processing of sequential inputs. While recurrent processing is valuable for time-varying inputs, it can also be useful for static inputs as recurrent networks have the ability to scale computation to varying difficulty levels by varying the number of recurrent iterations. This can be helpful when different problems from the same task family exhibit significant variations in complexity (consider mazes, for example). However, conventional RNNs typically struggle to automatically generalize across different difficulty levels, requiring retraining, fine-tuning, or human intervention to pick the number of recurrent iterations needed. This limitation hampers their practical utility in real-world scenarios, since the span of difficulty levels seen during training is finite.

---

[*]Equal contribution        [†]Work done while at UC San Diego

37th Conference on Neural Information Processing Systems (NeurIPS 2023).

Human reasoning, in contrast, is characterized by a highly adaptible use of computation - arbitrarily more computation can be used for more difficult tasks. Despite the evidence for such scaling in human computing there is limited work in the literature analyzing Adaptive RNNs (AdRNNs) that, on a sample-by-sample basis, automatically decide when to stop. Graves introduced Adaptive Computation Time (ACT), a mechanism to automatically stop computation, by learning to generate a scalar halting probability (Graves, 2016). However, similar to subsequent work (Banino et al., 2021), it was evaluated on either simple tasks such as parity checking, or language tasks. Few studies examine the training and evaluation of these AdRNNs on visual reasoning tasks, which pose unique challenges due to the redundant and high-dimensional nature of visual data. Furthermore, extrapolating to harder instances within-task has not been sufficiently studied, with Banino et al. (2021) evaluating such zero-shot performance only on the simple parity task.

In this paper, we study the problem of computation scaling in recurrent vision RNNs, with an emphasis on zero-shot extrapolation to harder/larger problems within the same task. The ability to handle unseen difficulty levels without fine-tuning, or human intervention, enables more robust and adaptable computer vision systems, which are crucial for real-world applications. We investigate the effectiveness of AdRNNs on two challenging, publically available, visual reasoning tasks based on curve tracing and route segmentation, namely PathFinder (introduced by Linsley et al. (2018)) and Mazes (introduced by Schwarzschild et al. (2021)).

The contributions of this work are as follows:

- We combine Convolutional RNNs with an adaptive computation method of Graves (2016) producing Adaptive ConvRNNs (AdRNNs) that are capable of learning a downstream task simultaneously while also learning to scale their computational steps as per input image/task requirements.
- We introduce LocRNN, a high performing recurrent architecture inspired by prior computational models of recurrence in biological vision.
- We show that AdRNNs learn to dynamically halt processing early (or late) to solve easier (or harder) problems when the train- and test-difficulty levels are matched on complex visual reasoning problems inspired by stimuli used in prior cognitive science research.
- During test time we introduce a previously unseen harder difficulty level. We evaluate AdRNNs on these new difficulty levels and show that they zero-shot generalize to more difficult problem settings not shown during training by dynamically increasing the number of recurrent iterations at test time well beyond the number of recurrent steps used during training.

## 2 Related Work

Our work is relevant to the visual routines literature introduced by Ullman (1984) and further reviewed elaborately by Roelfsema et al. (2000). The core idea of visual routines that make it relevant to our studied question of task extrapolation is the flexible sequencing of elemental operations resulting in a dynamic computational graph, which make RNNs a natural approach to solve such tasks.

Prior research has proposed mechanisms for RNNs that learn the amount of recurrent computational steps required. Following are two particularly relevant attempts in prior literature in this area: Graves (2016) developed the Adaptive Computation Time (ACT) method to train RNNs on Natural Language Processing tasks with a sigmoidal halting unit that determines when to halt recurrent processing, and Banino et al. (2021) extend upon this pioneering work by taking a probabilistic approach to halting and introducing a geometric prior-based specification of computational budget. However, neither of these studies applied adaptive computation to vision RNNs (that are significantly trickier to optimize). Also the prior studies did not attempt to examine computation scaling under zero-shot generalization to harder difficulty levels. Additionally, Eyzaguirre and Soto (2020) developed a version of ACT applied to visual reasoning problems, however, their observations on the CLEVR dataset (in which the issues related to object diversity are highlighted by Kim et al. (2018)) do not explore generalization to new difficulty levels which is central to our results.

In terms of generalization from easy to hard vision problems, our work is relevant to (Schwarzschild et al., 2021) and (Bansal et al., 2022) where the authors evaluate the application of recurrent neural networks to generalize from easier to harder problems. Our work extends and differs from their

intriguing study of recurrence in task extrapolation in the following ways: 1) For recurrent networks used in their study, human intervention is required to specify the number of recurrent computational steps during the testing phase. 2) Their work explores sequential task extrapolation in general with abstract problems such as Prefix Sum and solving Chess Puzzles while our work extends it to particularly focus on extrapolation in visual task learning. 3) We present evaluation on the Pathfinder challenge (Linsley et al., 2018), a relatively more large-scale visual reasoning task shown to be very challenging (Tay et al., 2020), the design of which dates back to (Jolicoeur et al., 1986). 4) Schwarzschild et al. (2021) and Bansal et al. (2022) implement only the most straightforward form of recurrence realized by weight-tying, i.e., they only evaluate ResNets with weight sharing across residual blocks (albeit with a novel training scheme in the follow up study (Bansal et al., 2022)). In addition to such recurrent ResNets, we present analyses with highly sophisticated recurrent architectures specialized for recurrent image processing. 5) We introduce LocRNN, a high performing recurrent architecture based on prior computational models of cortical recurrence.

On the design of recurrent architectures, our work is loosely related to (Eigen et al., 2013), (Pinheiro and Collobert, 2014), (Veerabadran and de Sa, 2020) and (Liao and Poggio, 2016) which discuss the role of weight sharing in feedforward networks to produce recurrent processing. We are interested, however, in designing new specialized recurrent architectures that play a role both in human and machine vision. While there exist more such recurrent architectures informed by neuroscience such as Linsley et al. (2018); Nayebi et al. (2022), we find these architectures to be quite difficult to interpret and unstable to train (based on in-difficulty performance evaluation included in the Supplementary). Our proposed LocRNN architecture in comparison is an elegant and easy-to-interpret implementation of recurrence that is stable to train. Our work is also relevant to prior work on modeling speed-accuracy tradeoffs observed during visual perception (Spoerer et al., 2020).

## 3 Datasets

For evaluating the ability of various models in exhibiting task extrapolation, we curate two challenging visual reasoning tasks, Mazes and PathFinder, with instances at multiple parametric difficulty levels. Both tasks involve the visual routines of marking and curve tracing (Ullman, 1984). These datasets are inspired by prior visual psychophysics research where such tasks were used abundantly to estimate the cognitive and neural underpinnings of sequential visual processing, such as incremental grouping, structure and preference of lateral connections, etc. (Jolicoeur and Ingleton, 1991; Ullman, 1984; Li et al., 2006; Roelfsema, 2006). In the following subsections we describe the specifics of our tasks.

### 3.1 PathFinder challenge – curve tracing

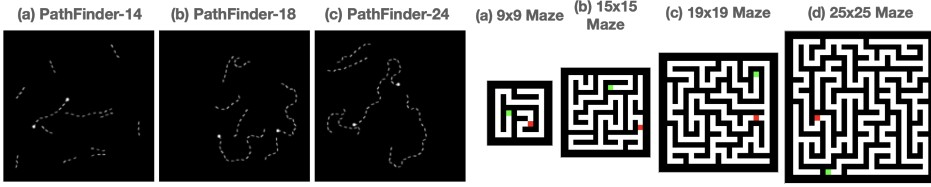

Figure 1: Representative examples from PathFinder and Mazes datasets. Left: (a) Positive PathFinder-9 (b) Negative PathFinder-18 (c) Positive PathFinder-24; Right: (a) 9×9 mazes, (b) 15×15 mazes, (a) 19×19 mazes, (a) 25×25 mazes

**Task description:** In the PathFinder task introduced by Linsley et al. (2018), models are trained to identify whether two circular disks in an input stimulus form the two ends of a locally connected path made up of small "segments". Each image consists of two main long connected paths $P_0$ and $P_1$ made up of locally aligned segments as well as shorter distractor paths. Each image also contains two circular disks which are placed at two of the 4 possible endpoints of $P_0$ and $P_1$. Images that contain a disk on both ends of the same path are classified as positive, and those containing a disk on endpoints of different paths are classified as negative. Examples are shown in Figure 1.

**Difficulty levels:** Pathfinder is designed at different difficulty levels parameterized by the length (number of segments) of the paths $P_0$ and $P_1$ mentioned above. The easiest version uses paths that are 9 segments long (PathFinder-9), while the medium and hard versions contain paths that are

14 (PathFinder-14) and 18 (PathFinder-18) segments long respectively (see example images in the Appendix). This dataset consists of a total of 800,000 RGB images at each difficulty level, each with a spatial resolution of $160 \times 160$ pixels. There are an equal number of positive and negative instances at each difficulty level. We use 700,000 images for training and 100,000 images as the test set from each difficulty level. We combined these datasets to create a more challenging dataset with varying levels of difficulty, which we refer to as PathFinder-Mixed. To evaluate the zero-shot difficulty extrapolation on PathFinder in Sec. 5.2 we generated 100,000 images each with contour lengths 21 and 24 respectively; we call these PathFinder-21 and PathFinder-24.

**Evaluation criteria:** Since this is a classification challenge, we use accuracy, i.e. *% correct on test-images* as the evaluation metric to rank model performance on PathFinder. Model architectures receive an input image and process it via a stack of standard convolution/recurrent-convolution layers followed by a classification two-class readout. Since this is a binary classification challenge with balanced classes, chance performance is 50% for random predictions.

### 3.2 Mazes challenge - route segmentation

**Task description:** Human beings are adept at solving mazes, a task that requires application of a similar serial grouping operation like PathFinder in order to discover connectivity from a starting point to the final destination of the maze amidst blocking walls. For evaluating model performance on solving mazes of varying difficulty, we use the publicly available version of the Mazes challenge developed by Schwarzschild et al. (2021). They implemented this Mazes challenge as a binary segmentation problem where models take $N \times N$ images of square-shaped mazes as input with three channels (RGB) with the start position, end position, permissible regions and impermissible regions marked in the image. The output produced by models is a binary segmentation of the route discovered by the model from the start to the end position.

**Difficulty levels:** The Mazes challenge has been designed at several difficulty levels, each difficulty level is parameterized by the size of the square grid that the maze is designed into. We use maze datasets of grid sizes 9×9 and 15×15 for training. Each dataset consists of 50,000 training images and 10,000 test images that are guaranteed to not overlap. The spatial resolution of 9×9 maze images is $24 \times 24$ pixels and that of 15×15 maze images is $36 \times 36$ pixels. Similar to PathFinder-Mixed, we combine 9×9 and 15×15 grid sizes to form Mazes-Mixed. We also constructed larger mazes with grid sizes $19 \times 19$ (with spatial resolution $44 \times 44$ pixels) and $25 \times 25$ (with spatial resolution $56 \times 56$ pixels) to evaluate a model's ability to extrapolate to difficulties not seen during training. As above with PathFinder's extrapolation evaluation, these novel difficulty mazes were only used during testing and were not in any form used in the training or hyperparameter optimization process.

**Evaluation criteria:** Mazes is a segmentation challenge and hence, one could potentially consider partially correct routes during evaluation (for example with average of per-pixel accuracy). However this is less strict than giving each image a single binary score reflecting if *all* pixels are labeled correctly. Evaluation criteria for mazes is hence the total *% of test-set mazes completely accurately solved* at a given difficulty level.

## 4 Model architectures and training

### 4.1 Implementations of adaptive computations evaluated on task extrapolation

In this section, we describe our choice of recurrent architecture designs that we use to study the behavior of adaptive computation and task extrapolation in deep learning. We process images (denoted as $\mathbf{X} \in \mathbb{R}^{c,h,w}$) in three stages that are common to all models we evaluate in this study. These three stages are as follows: (1) an input convolution layer (denoted as $\texttt{input}(.)$) that operates on the image directly.

$$\mathbf{h_0} = \texttt{ReLU}(\texttt{input}(\mathbf{X})) \tag{1}$$

The output from this stage $\mathbf{h_0} \in \mathbb{R}^{d,h,w}$ is fed as input to the following recurrent block in (2).

$$\mathbf{h_t} = \mathbf{r}(\mathbf{h_{t-1}}, \mathbf{h_0}), t \in [1, t_{train}] \tag{2}$$

where $\mathbf{r}(.)$ is the recurrent block. $t_{train}$ is a training hyperparameter indicating the maximum number of timesteps for unrolling $\mathbf{r}(.)$ during training. This block consists of a sequence of convolution

layers applied in an iterative manner for any arbitrary number of timesteps. This is the sub-part of our model that is capable of performing adaptive computations. (The specific architecture of these convolution blocks constitute different implementations of recurrent operations described further below.) (3) a readout layer containing a block of convolution and pooling operations that produce the desired output from our network.

$$\hat{\mathbf{y}}_{\mathbf{t}} = \texttt{readout}(\mathbf{h_t}), t \in [1, t_{train}] \tag{3}$$

While we keep the input and readout layer architectures the same for all models we evaluate, their intermediate recurrent blocks have different architectures (corresponding to their respective recurrent cells). We explore three different implementations of recurrent computations which are used in the second, recurrent block of models. Our first choice as the intermediate feature processing block consists of a residual network with weight tying across all layers; we refer to this model as R-ResNet-30. Next, we study the performance of the following specialized convolutional recurrent units from prior work: horizontal convolutional GRU (hConvGRU) and its stable variant (Linsley et al., 2018, 2020) and a convolutional Gated Recurrent Unit (ConvGRU) (Ballas et al., 2015) with LayerNorm (ConvGRU does not converge in the absence of LayerNorm). Third, we design a novel recurrent cell based on prior computational models of cortical recurrent processing (Li et al., 2006); this model equips the biologically inspired design choices of long-range lateral interactions, gating, and a separate population of interneurons. This model is referred to as LocRNN and is described in the following Sec. 4.3. Importantly all models are matched for trainable parameters.

## 4.2 Combining ConvRNNs with Adaptive Computation Time (ACT)

The central theme of our work is to show that RNNs can flexibly adapt (or scale) their computation according to input requirements. We achieve this ability by combining ConvRNNs with an adaptive computation mechanism based on Graves (2016) called Adaptive Computation Time (ACT). A difference between our work and ACT is that our visual reasoning task involves static inputs (i.e. sequence of length 1) whereas Graves (2016) deals with variable-length sequences and learns adaptive processing of each token. Owing to this difference, our halting mechanism is similar to ACT applied to a 1-token input sequence.

The key idea of ACT is to introduce a separate "halting mechanism" that learns to control the number of recurrent computation steps dynamically, conditioned on each input example's processing. In addition to producing the next recurrent state, an RNN equipped with ACT also produces a scalar value called the "halting score" for each computation step. In addition to the next hidden state computed using Equation 2, we generate a scalar halting score $p_t$ at each step using a learnt convolution layer (`halt_conv(.)`) that is shared across timesteps:

$$p_t = \sigma(\texttt{max\_pool}(\texttt{halt\_conv}(\mathbf{h_{t-1}}))) \tag{4}$$

The RNN treats the cumulative sum of the halting scores up to timestep $t$ ($P_t$ described below) as a quantity used to determine whether processing is terminated at that timestep.

$$P_t = \sum_{t'=1}^{t} p_{t'}$$

The RNN keeps track of the accumulated halting scores and checks if the cumulative sum of the halting scores at each step reaches a predefined threshold $(1 - \epsilon)$. If the threshold is reached, the computation stops (i.e., $p_t = 0 \ \forall t > t_{halt}$) where the cumulative halting score threshold is reached at timestep $t_{halt} = \min\{t : P_t >= (1 - \epsilon)\}$. The adaptive hidden state is then computed as a weighted average of the hidden states up to $t_{halt}$, scaled by the halting scores at each time step. The readout is then applied to this final adaptive hidden state to produce the ACT task prediction, $\hat{\mathbf{y}}_{act}$:

$$\mathbf{h}_{act} = \sum_{t=1}^{t_{halt}} p_t \cdot \mathbf{h_t} \tag{5}$$

$$\hat{\mathbf{y}}_{act} = \texttt{readout}(\mathbf{h}_{act}) \tag{6}$$

The abovementioned ConvRNN combined with ACT is trained to optimize both the downstream task loss ($||\mathbf{y} - \hat{\mathbf{y}}_{act}||_p$) and an auxiliary 'ponder cost' corresponding to the halting mechanism

that encourages models to use the fewest number of recurrent step to process an input example. Ponder cost is computed as the cumulative halting score until timestep $t_{halt} - 1$, maximizing this term encourages (or minimizing its negative as shown below) completing the task as quickly (with as few recurrent steps) as possible. The following final objective function is optimized to train the full model containing ConvRNNs with ACT where $\tau$ is a hyperparameter.

$$\mathcal{L} = \sum_{i=0}^{i=||\mathcal{D}||} \frac{1}{||\mathcal{D}||} ||\mathbf{y}^i - \hat{\mathbf{y}}^i_{act}||_p - \tau \sum_{t=1}^{t^i_{halt}-1} p^i_t \tag{7}$$

### 4.3 Formulation of LocRNN

We note that prior work has explored the development of recurrent architectures tailored for vision. Here, we introduce a similar but highly expressive recurrent architecture designed based on a computational model of iterative contour processing in primate vision from Li (1998). This model is an ODE-based computational model of interactions between cortical columns in primate area V1 mediated by "lateral connections" local to a cortical area.

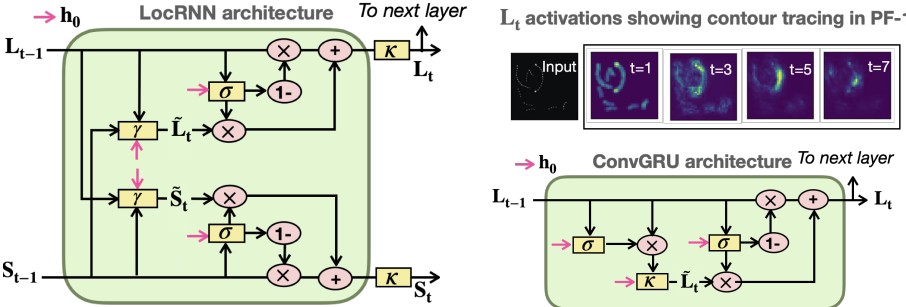

Figure 2: Contrasting the architectures of LocRNN (left) and ConvGRU (bottom-right). $\mathbf{h_0}$ is shown as a magenta arrow and does not change across timesteps. As illustrated also in this figure, LocRNN's interneuron activations ($\mathbf{S_t}$) are not passed to the next layer. ConvGRU on the other hand uses a single uniform neural population. (Top-right) shows a visualization of LocRNN's $\mathbf{L_t}$ activations as they perform contour tracing to solve an input image from PathFinder-14 (PF-14) (visualization of all timesteps available in Supplementary Fig. 8).

By discretizing the continuous-form ODE dynamics of processing units from Li (1998), we arrived at an interpretable and powerful set of dynamics for "LocRNN". As in ConvGRU, the effective receptive field of LocRNN output neurons increases linearly with recurrent timesteps.

Unlike ConvGRU, the LocRNN's hidden state is composed of two neural populations, $\mathbf{L}$ and $\mathbf{S}$; the activity in these populations are referred to as $\mathbf{L_t}$ and $\mathbf{S_t}$ at timestep $t$ respectively. These two populations are motivated by the $x$ (excitatory) and $y$ (inhibitory) neurons in Li (1998). However we found that restricting the signs of their weights (to reflect exclusively excitatory and inhibitory connections) led to less stable behavior. On the other hand, retaining the interneuron property (local computation not projecting out for downstream processing) of the $\mathbf{S}$ population performed better than using a single uniform population. The following equations illustrate the working of ACT and how the readout is applied in LocRNN.

$$p_t = \sigma(\texttt{max\_pool}(\texttt{halt\_conv}(\mathbf{L_{t-1}}))) \tag{8}$$

$$\hat{\mathbf{y}}_{act} = \texttt{readout}\left(\sum_{t=1}^{t_{halt}} p_t \cdot \mathbf{L_t}\right) \tag{9}$$

Initially, both $\mathbf{L_0}$ and $\mathbf{S_0}$ populations are set to a tensor of zeros with the same shape as $\mathbf{h_0}$, a $4d$ tensor of shape (batch_size$\times$channels$\times$height$\times$width).
$L$ and $S$ update gates $\mathbf{G^L_t}$ and $\mathbf{G^S_t}$ (same shape as $\mathbf{L}$ and $\mathbf{S}$) are computed as functions of the input and current hidden states $\mathbf{L_{t-1}}$ and $\mathbf{S_{t-1}}$ using 1x1 convolutions $\mathbf{U_*}$.

$$\mathbf{G^L_t} = \sigma(LN(\mathbf{U_L} * \mathbf{h_0}) + LN(\mathbf{U_{L \to L}} * \mathbf{L_{t-1}})) \tag{10}$$

$$\mathbf{G_t^S} = \sigma(LN(\mathbf{U_S} * \mathbf{h_0}) + LN(\mathbf{U_{S \to S}} * \mathbf{S_{t-1}})) \tag{11}$$

Each of the 4 types of lateral connections are modeled by convolution kernels $\mathbf{W_{L \to L}}, \mathbf{W_{L \to S}}, \mathbf{W_{S \to S}}$, and $\mathbf{W_{S \to L}}$ respectively of shape $d \times d \times k \times k$ where $d$ is the dimensionality of the hidden state and $k$ represents the kernel spatial size.

$$\tilde{\mathbf{L}}_\mathbf{t} = \gamma(\mathbf{W_L} * \mathbf{h_0} + \mathbf{W_{L \to L}} * \mathbf{L_{t-1}} + \mathbf{W_{S \to L}} * \mathbf{S_{t-1}}) \tag{12}$$

$$\tilde{\mathbf{S}}_\mathbf{t} = \gamma(\mathbf{W_S} * \mathbf{h_0} + \mathbf{W_{L \to S}} * \mathbf{L_{t-1}} + \mathbf{W_{S \to S}} * \mathbf{S_{t-1}}) \tag{13}$$

Once the long-range lateral influences are computed and stored in $\tilde{\mathbf{L}}_\mathbf{t}$ and $\tilde{\mathbf{S}}_\mathbf{t}$, these are mixed with the previous hidden states using the gates computed in Eq. 10 and 11. These hidden states are then passed on to subsequent recurrent iterations where even longer-range interactions occur (as time increases).

$$\mathbf{L_t} = \kappa(LN(\mathbf{G_t^L} \odot \tilde{\mathbf{L}}_\mathbf{t} + (1 - \mathbf{G_t^L}) \odot \mathbf{L_{t-1}})) \tag{14}$$

$$\mathbf{S_t} = \kappa(LN(\mathbf{G_t^S} \odot \tilde{\mathbf{S}}_\mathbf{t} + (1 - \mathbf{G_t^S}) \odot \mathbf{S_{t-1}})) \tag{15}$$

In the above equations, $LN()$ stands for Layer Normalization (Ba et al., 2016), and the nonlinearities $\gamma$ and $\kappa$ are both set to ReLU. One of the differences between LocRNN and ConvGRU (which we also evaluate) is the presence of the interneuron $\mathbf{S}$ population in the former.

## 5 Results

### 5.1 Adaptive RNNs scale their computation as a function of input difficulty

In this section, we report our observations from training AdRNNs on a mixture of difficulty levels from PathFinder and Mazes & evaluating them on a held-out set of images from the same mixture of difficulty levels of PathFinder and Mazes. We created one training dataset for each of these challenges by combining input samples from multiple "easy" difficulty levels. Combining input samples from multiple difficulty levels increases the diversity of computational requirements during training and helps in developing models that do not degenerate to using the same computational steps for all input samples.

As described in Section 3, we created a PathFinder training set of images and corresponding labels by sampling an equal number of images from three difficulty levels: PathFinder-9, PathFinder-14, and PathFinder-18. Similarly, we created a Mazes training set of images and ground-truth segmentation labels by sampling an equal number of images from two difficulty levels: 9x9 mazes and 15x15 mazes.

On held-out test sets that matched the difficulty level of the above-described training data, we tested whether models were able to scale their recurrent computational steps as a function of the input difficulty level. We show the results from this evaluation in Table. 1. **We observed that both the sophisticated AdRNNs tested (ConvGRU and LocRNN) were able to generalize to the held-out set.** We also observed that variants of horizontal GRU (Linsley et al., 2018) and Linsley et al. (2020) models generalized to the (within-difficulty) held-out set without using ACT. The simpler recurrent network without gating, weight-tied R-ResNet-30, was unable to learn on PathFinder and on Mazes highlighting the importance of specialized operations such as gating and backpropagation through time. The presence of skip connections between R-ResNet-30's recurrent blocks could still not match the expressivity of the specialized RNNs.

If ACT is working as expected, we must observe that AdRNNs that learn the task dynamically use less compute to solve easy examples and more compute (more recurrent iterations) to solve harder examples. To check for this trend, we analyzed the number of steps chosen by the model before halting for each example from the validation sets of PathFinder and Mazes; with varying contour lengths and maze sizes respectively. These results are shown by the cool colors in Figure 3 for PathFinder and in the Supplementary for Mazes. As is clearly observable from the trend in these Figures, **examples that we consider as harder (longer contours in PathFinder and larger mazes) are assigned a higher number of recurrent computation steps by ACT than easy examples.** Hence we show that AdRNNs obtained by combining ConvRNNs with ACT training are capable of learning both PathFinder and Mazes in addition to learning to adapt their recurrent computational steps as a function of input example difficulty. To the best of our knowledge, we are the first to show the above result for visual tasks inspired by stimuli used in prior cognitive science research.

|  | PathFinder-Mixed (%) | Mazes-Mixed (%) |
|---|---|---|
| ResNet-30 | 50.41 | 0.0 |
| R-ResNet-30 (ACT) | 49.37 | 0.0 |
| Linsley et al. (2020) | 50.0 | 78.33 |
| hConvGRU $t_{inference} = t_{train}$ | 89.66 | 99.69 |
| hConvGRU (stable halting) | 89.65 | 99.54 |
| ConvGRU (ACT) | 95.26 | 98.4 |
| LocRNN (ACT) (ours) | 97.13 | 98.4 |

Table 1: Accuracies (%) ↑ of models on the two visual reasoning tasks. Chance performance is 50% for pathfinder and 0% for Mazes.

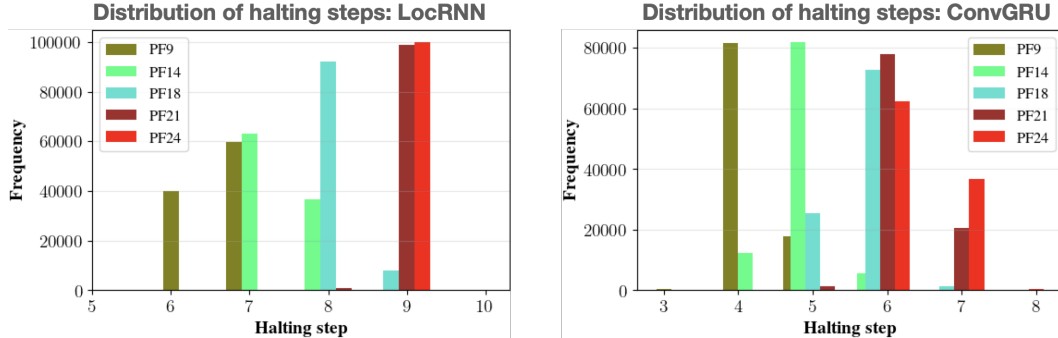

Figure 3: Distributions of halting steps across samples in each validation set of PathFinder. Computation appears to scale to match difficulties of the datasets for both LocRNN (left) and ConvGRU (right) models. 9-length contours typically halt after 4-6 steps, while 24-length contours can take up to 9 steps. The red bars show the distribution in the extrapolation datasets.

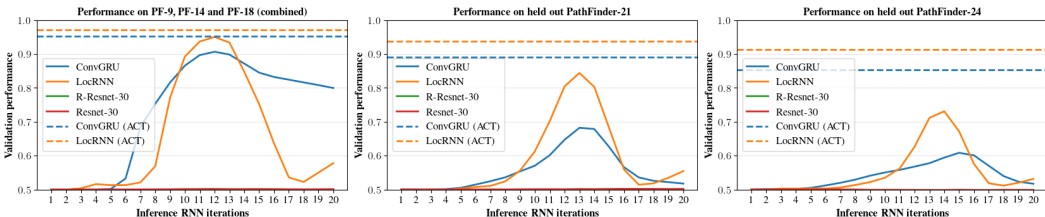

Figure 4: Vanilla RNNs (solid lines) and AdRNNs (dashed) trained on 12 iterations (a) within-difficulty (b) extrapolating to PathFinder-21 (c) extrapolating to PathFinder-24

## 5.2 Adaptive RNNs generalize to novel difficulty levels by scaling their computation

In typical circumstances where RNNs are expected to solve instances that are more difficult, human intervention in the form of specification of the number of recurrent iterations is commonplace (Schwarzschild et al., 2022). This approach is both laborious and expensive to perform as it requires training and/or evaluation on new test sets with various settings of the number of recurrent timesteps. Additionally, this process needs to be repeated at every occasion of new incoming test data that is potentially of increased difficulty. In the previous section, we showed the ability of AdRNNs in learning to solve PathFinder and Mazes while also demonstrating that they assign computation as a function of input difficulty on a test-set with matched task difficulty level as the training set. Here we are interested in testing whether AdRNNs are able to extrapolate (by using more recurrent computational steps compared to training) in complex reasoning tasks. We constructed two additional datasets each in the PathFinder and Mazes families to test these adaptive models' ability to extrapolate, as mentioned in Section 3. While AdRNNs are trained on the training difficulty levels for a maximum of $t_{train}$ iterations, **we evaluated them on these new (more difficult) datasets at inference with a maximum of $t_{inference} \geq t_{train}$ recurrent iterations where we refer to AdRNNs as operating in their extrapolation phase.** We report the results from this evaluation in Table. 2 and in Fig. 4.

|  | PathFinder-21 (%) | PathFinder-24 (%) | Mazes-19 (%) | Mazes-25 (%) |
|---|---|---|---|---|
| ResNet-30 | 50.0 | 50.0 | 0. | 0. |
| R-ResNet-30 (ACT) | 50.0 | 50.0 | 0. | 0. |
| Linsley et al. (2020) | 50.0 | 50.0 | 2.93 | 0.01 |
| hConvGRU $t_{inference} = t_{train}$ * | 64.21 | 58.35 | $16.2 \pm 2.66$ | $5.29 \pm 0.26$ |
| hConvGRU (stable halting) | 50.0 | 50.0 | $50.26 \pm 6.04$ | $21.36 \pm 2.82$ |
| ConvGRU (ACT) | $82.63 \pm 4.84$ | $74.14 \pm 6.52$ | $75.1 \pm 11.96$ | $46.93 \pm 4.2$ |
| LocRNN (ACT) (ours) | $\mathbf{92.89 \pm 0.9}$ | $\mathbf{85.81 \pm 5.57}$ | $\mathbf{86.83 \pm 2.94}$ | $\mathbf{49.99 \pm 4.48}$ |

Table 2: Accuracies (Mean % $\pm$ SEM) $\uparrow$ on extrapolation datasets. Chance performance is 50% for pathfinder and 0% for Mazes.
*hConvGRU training converged only on 1 out of 3 seeds (which were chosen randomly for all models here) on PathFinder, corresponding results above show this converged model's performance.

**Adaptive RNNs generalize to novel difficulty levels by scaling their computation** Our evaluation of AdRNNs trained with ACT on novel harder difficulty levels shows that as expected, AdRNNs have learned to use the optimal number of recurrent computational steps required for achieving strong generalization to the novel difficulty levels on a per-image level. On PathFinder, as seen in Figure 3, most instances from PathFinder-21 and PathFinder-24 take 9 steps in LocRNN and up to 7 steps in ConvGRU, higher than the number of steps used for easy training difficulty examples. In Fig 5 we visualize the relationship between mazes' difficulty and the number of recurrent iterations used by ACT (on a per-instance level). We observe that maze difficulty (length of the ground-truth route in pixels) and the number of ACT iterations are strongly positively correlated. Notably, AdRNNs trained with ACT choose to make $t_{halt}$ during inference on longer mazes greater than $t_{halt}$ used on the shorter training mazes.

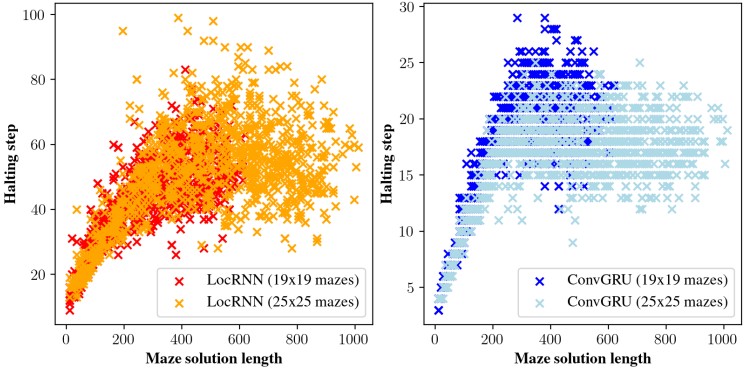

Figure 5: Relationship between halting step and difficulty level of Mazes for the extrapolation evaluation . Here we visualize the relationship between the halting step and solution length of mazes at a per-instance level. We note a strong positive correlation between the difficulty level (length of maze solution segmentation) and halting step used by LocRNN (left) and ConvGRU (right) AdRNNs.

### 5.3 AdRNNs outperform halting in hConvGRU based on stability of hidden-state dynamics

Using stability in hidden-state space as an alternative way to perform halting during extrapolation, we evaluate hConvGRU on extrapolation. For this method, we halted processing in hConvGRU when the mean absolute difference between two subsequent states ($\|\mathbf{h_t} - \mathbf{h_{t-1}}\|$) reduced to less than a tenth of the difference between the first two states (heuristic for stability). This method referred as hConvGRU (stable halting) in Tables 1 and 2 shows differences in generalization across our datasets. **AdRNNs perform better than hConvGRU (stable halting) on both datasets**; especially, stable halting completely fails to show generalization to PathFinder-21 and -24. We hypothesize this method to work suboptimally for the following reasons:

- Stability-based halting requires defining a hand-engineered heuristic on how much change in the output is considered small enough to halt. In typical segmentation tasks like Mazes, the network output for the initial few timesteps is highly stable in making nonsensical

predictions (predicting all pixels as the negative class) and thus, heuristics need to identify an inflection point in the output trajectory where meaningful predictions start to emerge and stabilize. In the absence of ground truth information, one cannot pick a heuristic that generalizes to unseen data.

- Learnable halting makes less assumptions about the hidden state's properties, and hence doesn't enforce hard constraints such as stability to be satisfied by training. Some, but not all RNNs, have stable hidden states wherein the network response stops changing after reaching an attractor. Linsley et al. (2020) argue that RNNs that are expressive have an intrinsic inability to learn stable hidden states. For RNNs to be stable, their hidden state transformation needs to model a contractive mapping (Miller and Hardt, 2018; Pascanu et al., 2013). That is, the recurrent transition function $F$ satisfies the following inequality: $||F(h_t) - F(h_{t-1})||_p < \lambda||h_t - h_{t-1}||_p$. RNNs with stable hidden states that satisfy the above inequality are quite difficult to train on challenging problems in practice. Even when stable models perform comparably to unstable models as in (Miller and Hardt, 2018), the authors show unstable models' advantages such as performance improvements in the short-time horizon and lesser vanishing gradient issues.

## 6 Conclusion

The advantage of using recurrent networks for processing static inputs adaptively, particularly in order to zero-shot generalize to new difficulty levels is understudied. In this work, we show that deep convolutional networks with intermediate recurrent blocks (ConvRNNs operating on image features) can be combined with an adaptive computation technique to learn to dynamically process different input examples based on a per-instance difficulty level. We combine ConvRNNs with a learnable halting mechanism that is based on Graves (2016) to produce AdRNNs. We evaluated diverse implementations of recurrence in increasing level of sophistication/complexity, R-ResNet-30 with weight-tying, ConvGRU and hConvGRU with gating, and LocRNN with two separate populations of horizontally connected units (one of which are interneurons) and gating on two challenging visual reasoning problems, PathFinder and Mazes, which are generated at various levels of difficulty. First, we showed that only the specialized RNNs (hConvGRU) and AdRNNs (LocRNN and ConvGRU) are capable of learning PathFinder and Mazes. Second, these AdRNNs trained with ACT are learning to dynamically use less (or more) recurrent computational steps for easy (or hard) PathFinder and Maze problems respectively as discussed in Section. 5.1. More interestingly when the difficulty level of the test set was increased relative to training difficulty levels, AdRNNs generalized to these harder instances in a zero-shot manner by allocating more recurrent computation than was ever used during training. They also outperform stability heuristics-based adaptive recurrent networks. Our work empirically shows this hypothesized advantage of using adaptive recurrent processing on static-image tasks for the first time to the best of our knowledge.

In addition to adaptively scaling computation according to the need, the AdRNNs can also solve the problems more efficiently, choosing to stop at earlier times than the non-adaptively trained RNNs (compare halting times in Fig 3 with performance curves in Fig 4) with human arbitrarily chosen training iterations. Thus, just as computation can be scaled to unseen problem difficulties without human intervention, the training iterations required can be (better) discovered automatically.

## 7 Limitations and Future Work

The current work has been only applied to static images, but recurrent networks are able to process time-varying input; future work will explore our models on video input. In addition, there appears to be a benefit of the LocRNN model over ConvGRU that deserves further study to determine the mechanism behind the benefit of the interneuron population.

For future work, we would like to further explore the benefits of training only on very small problem instances with the ACT mechanism automatically allowing for harder instances. This could greatly reduce latency and energy consumption both during training and inference, an increasingly relevant concern as the size, memory footprint, training time and inference latency of models grow drastically. Finally, we wish to explore how this approach might link to the human developmental literature, similar to earlier work by Elman (1993).

# 8 Acknowledgments & Funding disclosure

We thank Garrison W. Cottrell, Michael C. Mozer, Michael L. Iuzzolino, Pradeep Shenoy and Saurabh Singh for their feedback and insights on this work. This work is funded by: (1) NSF CRCNS Award No: 2208362, (2) Sony Faculty Innovation Award, and (3) Kavli Symposium Inspired Proposal and support from Social Sciences Computing Facility and Department of Cognitive Science at UC San Diego.

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

# Supplementary

## A   Training and implementation details

All architectures we evaluated within a task were matched in terms of the number of parameters. The input convolutional layer's kernel size is $7 \times 7$. Number of channels used by the model remains unchanged across layers and is determined per-model for matching overall number of trainable parameters across models. For PathFinder, we fix the number of channels to be 32 for LocRNN, 21 for hConvGRU and ConvGRU, and 64 for ResNet-30 with a filter size of 9x9 in the intermediate recurrent layers. For Mazes, we fix the number of kernels (d) to be 128 for LocRNN, ConvGRU, hConvGRU, and 100 for ResNet-30 & R-ResNet-30 and the kernel size is fixed to be 5x5. Our readout layers for classification (PathFinder) contain a global average pooling layer followed by a fully-connected layer with output dimensionality of 1 producing the classification logit. We apply binary cross-entropy loss on the logit to train models on PathFinder. On Mazes, we use a $1 \times 1$ convolution with 1 output channel to produce a binary segmentation map. We use pixel-wise binary cross-entropy to train models on Mazes. These readout layers are used uniformly for all architectures evaluated.

On Mazes training minibatch size is set to 64 images (and inference batch size of 50 images) and a learning rate schedule starting with warmup followed by step learning rate decay as indicated in Schwarzschild et al. (2021) for 50 total epochs of training. On PathFinder, we set the training minibatch size to 256 images and a constant learning rate of 1e-4 for all models for a total of 20 epochs of training. All models were trained on NVIDIA RTX A6000 GPUs and implemented using PyTorch.(Paszke et al., 2017).

## B   Instability of other baseline ConvRNNs

ConvRNN training is often faced with instability issues that lead to sensitivity with respect to random seeds or lack of convergence of models on downstream tasks. We tested a suite of ConvRNNs previously introduced that are similar to LocRNN on three difficulty levels of PathFinder (in-difficulty evaluation, i.e., training and testing on each difficulty level independently). This evaluation highlighted the above issue especially on the difficult levels of PathFinder where LocRNN was the only model which could converge to stable solutions across different random seeds unlike the other networks which performed at chance as shown below in Fig. 6.

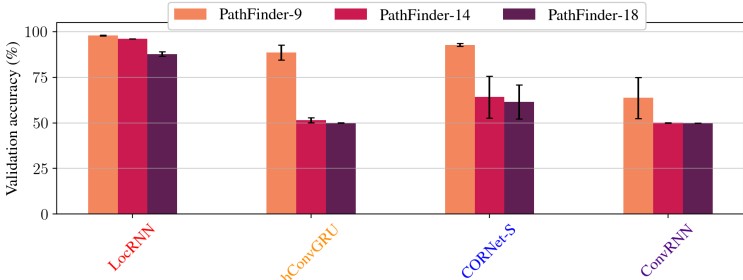

Figure 6: Performance of various ConvRNN models on PathFinder-9, PathFinder-14, and PathFinder-18.

## C   Input and output format for PathFinder and Mazes

Each example maze is an $n \times n$ RGB matrix, with colored squares indicating the start (green) and end (red) positions in the maze. The output is a binary matrix of size $n \times n$ with the segmented path indicating the maze solution. An example is shown in Figure 7 (top).

Each PathFinder example is an $n \times n$ binary matrix as shown in Figure 7 (bottom). The output for one sample is a pair of probabilities denoting which class the sample belongs to (negative, meaning

the disks are at the end of disconnected paths, or positive, meaning the disks are connected through the contour).

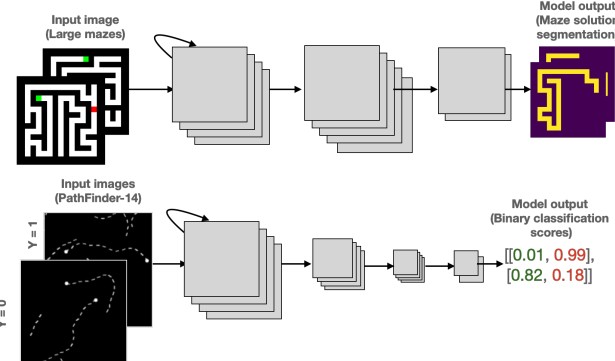

Figure 7: (Top) Example images from 11×11 Mazes processed by a model to produce the solution as a segmentation prediction. (Bottom) Example input images from PathFinder-14 processed by a classifier to produce binary classification output.

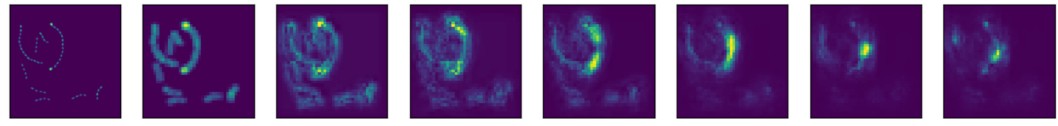

Figure 8: State activations $L_t$ of LocRNN for a PathFinder-14 example, clearly displaying the contour integration strategy used by LocRNN. The activation maps suggest that the contours are integrated from both endpoints, and a decision is taken based on whether they meet or not.

