# OpenReview forum: "Adaptive recurrent vision performs zero-shot computation scaling to unseen difficulty levels"
_NeurIPS.cc/2023/Conference — NeurIPS 2023 poster_

### Official Review · Reviewer_qD61 · 2023-06-26

**Soundness:** 3 good
**Presentation:** 1 poor
**Contribution:** 3 good
**Rating:** 5
**Confidence:** 4

**Summary:**

The paper introduces a combination of Convolutional Recurrent Neural Networks (ConvRNNs) with a learnable termination mechanism from Adaptive Recurrent Neural Networks (AdRNNs), with the purpose of solving complex, variable-difficulty vision tasks. The paper adapts to the purpose a ConvGRU architecture, or alternatively a “LocRNN” architecture based on a computational model of iterative contour processing in primate vision. Both architectures, together with a ResNet baseline, are tested on the “Pathfinder” and “Mazes” tasks, with a variety of task difficulties. Importantly, the authors show that the AdRNNs architecture are able to generalize to task difficulties greater than those seen during training, requiring a number of computation steps beyond those used in training.

**Strengths:**

The paper consists of an interesting and timely investigation into the hot topic of adaptive computation for problem solving.

While the authors are not the first to work on this, they appear to be the first to have demonstrated that RNN models can be effectively applied for vision tasks with adaptive difficulty, and generalize to unseen difficulty levels.

The work is clearly motivated, and the overall story and contribution is immediately apparent.

**Weaknesses:**

I believe that the following points need to be addressed for my assessment of the paper to be changed more positively:
* **[a]** Overall, the main contribution of the paper is the introduction of LocRNN, as all results, including the one on generalization to unseen difficulty levels for tasks, are downstream of LocRNN being a better architecture than alternatives. However, Section 4.3 appears crowded and hard to parse. I believe that the inclusion of figures illustrating the architectures and describing the differences between ConvRNN and LocRNN would greatly improve the presentation. Moreover, not much effort is spent intuitively justifying the reasons behind the improved performance of LocRNN over other ConvRNNs.
* **[b]** In Line 197, it is mentioned that the computational models of cortical recurrent processing are described in detail in Supp. Info, but there is no such section in the supplementary material. Please include this section, to again better explain the intuition behind the introduction of LocRNN.
* **[c]** There are small formatting issues. In Line 40, citation names are used without brackets even when they don’t naturally flow within the sentence.

**Questions:**

What is the computational model of cortical processing relevant to the introduction of LocRNN? What is the intuition behind LocRNN’s improved performance and stability over other ConvRNNs?

**Limitations:**

The authors seem to have fairly acknowledged the limitations of their work in their “Limitations” section.

---

> ### Author Rebuttal · Authors · 2023-08-10
>
> We take the PDEs proposed in (Li, 1998), which model the connections between V1 neurons in continuous time. We take these PDEs, apply Euler integration to them in order to convert them into discrete-time difference equations and implement them using convolutional Recurrent Neural Networks. The final RNN obtained from Li 1998 after Euler integration is what we refer to as LocRNN in our submission. We have contrasted the architectures between LocRNN and GRU in Fig 8.
>
> We will include the original PDEs and the derivations which we missed in the Supplementary as well as fix the formatting issues.
>
> References:
> Tay et al. (2020), Tay, Y., Dehghani, M., Abnar, S., Shen, Y., Bahri, D., Pham, P., ... & Metzler, D. (2020). Long range arena: A benchmark for efficient transformers. arXiv preprint arXiv:2011.04006.

---

> > ### Comment · Reviewer_qD61 · 2023-08-10
> > **Reply to the Authors**
> >
> > We thank the authors for having included an illustration of LocRNN in the rebuttal pdf, and having clarified the origins of the architecture. I consider these points necessary for my score to remain a tentative accept.

---

### Official Review · Reviewer_cX6r · 2023-07-05

**Soundness:** 3 good
**Presentation:** 3 good
**Contribution:** 1 poor
**Rating:** 4
**Confidence:** 4

**Summary:**

Authors combine Adaptive computation time (ACT) with convolutional recurrent neural networks to solve two tasks with which their generalization properties can be studied. These adaptive-timestep RNNs were found to halt quicker for easier problems while taking more steps for harder ones. This also leads to generalization to an unseen difficulty level.

**Strengths:**

The paper is well written and the results are convincing for the claim.
* Pathfinder task is known to be challenging (Tay et al., 2020) so high accuracy and generalization on that task is very interesting.
* Adaptive choosing of timesteps within a dataset is important & valuable for neuroscience and cognitive science research since it allows for identification of harder and easier stimuli.
* Inference-time timesteps being more than training is a strong indicator that the RNNs are generalizing.

**Weaknesses:**

While I found the work interesting, I found the paper to be severely lacking in terms of novelty. The paper also misses some related works ([1], [2]).

*  As mentioned in the paper, Bansal et al., (2022) have studied this with the maze task. Authors say "recurrent networks used in their study are not adaptive, human intervention is required to specify the number of recurrent computational steps by brute force during the testing phase" which I do not agree with - since their RNN stops changing its response after a while, that point can be used as to decide when to stop.
* [2] has studied this with Pathfinder task - even finding generalization to difficult levels.

Between Graves et al. (2016), Bansal et al. (2022) and [2] I do not see the novelty of this work.






1. Spoerer CJ, Kietzmann TC, Mehrer J, Charest I, Kriegeskorte N (2020) Recurrent neural networks can explain flexible trading of speed and accuracy in biological vision. PLOS Computational Biology 16(10): e1008215. https://doi.org/10.1371/journal.pcbi.1008215
2. Drew Linsley, Alekh Karkada Ashok, Lakshmi Narasimhan Govindarajan, Rex Liu, Thomas Serre (2020) Stable and expressive recurrent vision models. NeurIPS 2020

**Questions:**

1. I am curious to hear why the authors say RNNs used in Bansal et al., (2022) "are not adaptive, human intervention is required to specify the number of recurrent computational steps by brute force during the testing phase" since the RNN can be run till response stops changing.

2. What properties of the images makes the RNNs stop early vs late? For example, in Fig 4 some 19x19 mazes seem to need the timesteps as 25x25 mazes. Have you seen what causes this? Same for the other task. Can this lead to some interpretation/understanding of the RNNs and the solution they are implementing?

---

> ### Author Rebuttal · Authors · 2023-08-10
>
> "I am curious to hear why the authors say RNNs used in Bansal et al., (2022) "are not adaptive, human intervention is required to specify the number of recurrent computational steps by brute force during the testing phase" since the RNN can be run till response stops changing"
>
> 1. As this is an important issue relevant to several reviews, we have addressed this general issue  in our common response. We also tried the  Linsley et al 2020 algorithm with a threshold on state change and it performed very poorly on Mazes (see Table 3 in the Rebuttal).
>
> "What properties of the images makes the RNNs stop early vs late? For example, in Fig 4 some 19x19 mazes seem to need the timesteps as 25x25 mazes. Have you seen what causes this? Same for the other task. Can this lead to some interpretation/understanding of the RNNs and the solution they are implementing?"
>
> 2, The plot in Fig 4 plots halting step as a function of maze solution path length.  There is a clear linear relationship between solution path length (not maze size) for the shorter path lengths. For longer maze path lengths, the required computation saturates.

---

> > ### Comment · Reviewer_cX6r · 2023-08-18
> > **Reply to author rebuttal**
> >
> > Thank you the responses and the additional experiments.
> >
> > >"I am curious to hear why the authors say RNNs used in Bansal et al., (2022) "are not adaptive, human intervention is required to specify the number of recurrent computational steps by brute force during the testing phase" since the RNN can be run till response stops changing"
> > > **Response:** As this is an important issue relevant to several reviews, we have addressed this general issue in our common response. We also tried the Linsley et al 2020 algorithm with a threshold on state change and it performed very poorly on Mazes (see Table 3 in the Rebuttal).
> >
> > I understand that there are downsides to Bansal et al. (2022) and Linsley et al. (2020) but that still doesn't mean that they are not adaptive. I would encourage updating the manuscript with the downsides and remove the claim that they are not adaptive.
> >
> >
> > > "What properties of the images makes the RNNs stop early vs late? For example, in Fig 4 some 19x19 mazes seem to need the timesteps as 25x25 mazes. Have you seen what causes this? Same for the other task. Can this lead to some interpretation/understanding of the RNNs and the solution they are implementing?"
> > > **Response:** The plot in Fig 4 plots halting step as a function of maze solution path length. There is a clear linear relationship between solution path length (not maze size) for the shorter path lengths. For longer maze path lengths, the required computation saturates.
> >
> > As you note, the RNN seems to have captured a factor beyond path lengths. For me it would be interesting if you used your adaptive timestep method to provide interpretations of an RNN - which I think is novel and beyond what was already done by Bansal et al. (2022) and Linsley et al. (2020). For me, this would push the paper above the threshold for publication. As it stands, given that ACT is a known method and Bansal et al. (2022) & Linsley et al. (2020) have studied adaptive time for Mazes & Pathfinder respectively, I do not see the novelty in simply introducing ACT for Mazes and Pathfinder.
> >
> >
> > From reading the responses and the discussion, it seems to me that the main contribution is in fact the new RNN architecture (LocRNN) and not the training algorithm. However, it is still unclear to me why LocRNN is better than convGRU, hconvGRU, CORNet-S etc. Like the LocRNN model, hconvGRU and CORNet-S are also neuroscience-inspired. So, what is the critical novel circuitry in LocRNN that enables such great performance? For example, LocRNN has LayerNorm while convGRU, hconvGRU and CORNet-S don't seem to - how critical is this for superior performance of LocRNN? LocRNN has separate inhibitory and excitatory populations (which could explain superiority to vanilla convGRU) but so does hconvGRU - so where exactly does the improvement come from? I think a systematic analyses where the LocRNN's components are lesioned and explicitly contrasted to other neuroscience-inspired architecture is necessary to gauge the contributions of LocRNN architecture.
> >
> > To summarize :
> > 1. **Limited advancement in adaptive computation compared to previous methods**: Although a previous method was added during rebuttal, I am skeptical of the results since it was done in a very short time-frame and some experiments are missing. Bansal et al. (2022) showed strong generalization for the maze task and Linsley et al. (2020) for the pathfinder task - I am skeptical without more experiments that they won't work for each other. I think more extensive evaluation is needed before this method can be concluded to be better - `all three methods {ACT, Bansal et al. (2022), Linsley et al. (2020)}  X  all models {LocRNN, convGRU, hconvGRU, CORNet-S} X all tasks {maze, pathfinder}`. As it is currently, we do not know where the improvements are coming from.
> >
> > 2. **Interesting model but unclear where the contribution is**: LocRNN model is interesting and seems to be bringing in performance improvements. But since it is similar to previously known models, it is unclear as to what advancement in the model is causing this improvement.

---

> > > ### Author Response · Authors · 2023-08-20
> > > **Response to reviewer's response to author rebuttal**
> > >
> > > We thank you for your response to our author rebuttal. Please find below our response to your latest comments.
> > >
> > > - _“I understand that there are downsides to Bansal et al. (2022) and Linsley et al. (2020) but that still doesn't mean that they are not adaptive. I would encourage updating the manuscript with the downsides and remove the claim that they are not adaptive.”_
> > >
> > > We agree with the reviewer, we will update the manuscript removing the claim that they aren’t adaptive and specifically mention the downsides we have highlighted in our rebuttal.
> > > - _"As you note, the RNN seems to have captured a factor beyond path lengths. For me it would be interesting if you used your adaptive timestep method to provide interpretations of an RNN - which I think is novel and beyond what was already done by Bansal et al. (2022) and Linsley et al. (2020)."_
> > >
> > > Thank you for the interesting suggestion to explore using adaptive timesteps to provide an interpretation of RNNs.
> > > We agree with you, we believe studying different models’ dynamics at- and around the halting timestep will provide intuition on properties that further differentiate LocRNN from the baselines compared.
> > >
> > > We have been actively trying to understand the roles of the two neuron populations in LocRNN, and how they contribute to the empirical results obtained. Fig 7 in the supplementary shows state activations of the L population for a PathFinder example. While the L population lends itself to easy interpretation as performing contour integration, we have inspected the S population, which is harder to understand. We find that they clearly differ from the L activations, suggesting that they’re encoding complementary information, but we are continuing to discover what these interneuron-analogues might be encoding.
> > >
> > > - _"So, what is the critical novel circuitry in LocRNN that enables such great performance? For example, LocRNN has LayerNorm while convGRU, hconvGRU and CORNet-S don't seem to - how critical is this for superior performance of LocRNN? LocRNN has separate inhibitory and excitatory populations (which could explain superiority to vanilla convGRU) but so does hconvGRU - so where exactly does the improvement come from?_
> > >
> > > Thank you for your thorough review of our architectural differences with baselines we compared against. Please find our detailed response to this question above in our common response where we highlight our hypothesis on why LocRNN works better in comparison to our baselines. While both LocRNN and hConvGRU seem to be using two separate neural populations, they are implemented differently in these two architectures as highlighted in our common response.
> > >
> > > - _“LocRNN has LayerNorm while convGRU, hconvGRU and CORNet-S don't seem to - how critical is this for superior performance of LocRNN?”_
> > >
> > >     * First, normalization is an essential component of many recent high-performing architectures like LocRNN.
> > >     * ConvGRU converged on PathFinder and Mazes only when LayerNorm was included in its architecture. For fairness, we only report performance of ConvGRU with LayerNorm in our main submission and rebuttal. We will mention this clearly in our updated manuscript.
> > >     * LayerNorm was not added to hConvGRU and CORNet-S as they achieved normalization through batchnorm. hConvGRU and CORNet-S both have batch normalization layer(s) which is crucial for their performance and we don’t change their architecture for consistency with their published results
> > >
> > > - _"To summarize :_
> > >
> > > _Limited advancement in adaptive computation compared to previous methods: Although a previous method was added during rebuttal, I am skeptical of the results since it was done in a very short time-frame…"_
> > >
> > > We understand the concerns on the experiment quality given the short rebuttal timeframe. We are confident in our reported results which used open source implementations provided by the respective authors of hConvGRU, CORNet-S and Bansal, 2022. We shall release the code we used for our experiments on acceptance and include instructions & environment required to reproduce our results.
> > >
> > > - _“more extensive evaluation is needed before this method can be concluded to be better - all three methods {ACT, Bansal et al. (2022), Linsley et al. (2020)} X all models {LocRNN, convGRU, hconvGRU, CORNet-S} X all tasks”_
> > >
> > > We find the results we have obtained from searching part of the above space to strongly suggest the superiority of LocRNN, we are continually actively exploring the rest of this search space to add further strength to our analysis.
> > >
> > > - _“Interesting model but unclear where the contribution is: LocRNN model is interesting and seems to be bringing in performance improvements.”_
> > >
> > > We agree that the model is not fully understood yet (like many previously published high-performing models and training techniques like Batch Normalization), but believe that it will be exciting to the NeurIPS community and thus catalyze further exploration.

---

> > > > ### Comment · Reviewer_cX6r · 2023-08-21
> > > >
> > > > Thank you for the comments. However, I still think the evaluations are severely limited and therefore I will set my final evaluation at 4 (3 $\rightarrow$ 4).

---

### Official Review · Reviewer_1Kd6 · 2023-07-08

**Soundness:** 3 good
**Presentation:** 2 fair
**Contribution:** 4 excellent
**Rating:** 5
**Confidence:** 3

**Summary:**

The manuscript presents adaptive recurrent networks for processing of static images. The proposed approach augments convolutional recurrent neural networks with the adaptive computation time mechanism, in which the RNN at each step computes an additional halting unit, the value of which is used to determine when to stop the iterative computation. This is shown to yield recurrent vision models that adapt their computational budget to scale with the difficulty level of tasks that require serial grouping operations, such as determining whether two points are located on opposite ends of the same path or segmenting the solution route of a maze. The experimental section also shows that adaptive recurrent networks with gating mechanisms successfully scale to solve the tasks at higher difficulty levels than seen during training.

**Strengths:**

The proposed approach is novel and well-motivated. The experimental results confirm that the proposed models are able to extrapolate the learned knowledge beyond the training distribution to solve more difficult versions of the two tasks. This is quite impressive and I'm not aware of similar studies.

**Weaknesses:**

The write-up is a bit difficult to follow:

1. It seems to me that some important details of the original ACT [1] halting mechanism are omitted in this manuscript or the proposed mechanism is a modified version, which should be clearly stated. I might be mistaken and only the formulation or notation is different. If so, please let me know.
2. Both $p_t$ and $P_{t'}$ are not really probabilities, as I understand it. You could maybe simply call $P_{t'}$ a quantity that, if it reaches a predefined threshold, halts computation.
3. $\phi$ in Equation 4 is not defined
4. It should be stated right after the first occurrence that LN stands for layer normalization.

I think some details for reproducibility are missing or unclear:

5. The sentence in line 188 states that *four* choices are explored, then in 191 it is stated that *three* different recurrent implementations are explored. As far as I understood, the first of the four choices is a non-recurrent ResNet-30. It is not clearly explained how this model is implemented as Equation 2 only covers recurrent architectures.
6. The exact architectures including hyperparameters (e.g. number of kernels $d$ and their sizes) of the recurrent models are not provided.
7. Why is R-ResNet-30 equipped with ACT in Table 1, but not in Table2?

I think the experimental setup should take the following into consideration:

8. For completeness, I think is would make sense to add a few more non-recurrent baselines. It seems intuitive that the considered tasks require multi-step reasoning, but it'd be important to show this empirically, e.g. by using a small vision transformer. I would be surprised if a transformer would completely fail on the task, given the fairly large number of training samples. I can, however, imagine that the proposed architecture performs much better on unseen difficulty levels. Showing this empirically would help in establishing AdRNNs as an important alternatives to current vision models.
9. Accuracies are not reported as mean and standard deviation over multiple runs with different random seeds. This is very problematic, since with some tasks and architectures, single runs may underperform or not converge to any meaningful solution at all, especially with binary success measures (such as exactly correct maze segmentation). Aggregating metrics over several runs (e.g. 5) would help the reader interpret how significant a different in performance really is (e.g. is it more than 1 standard deviation?)

Regarding references:

10. I think the correct citation for *convolutional* GRUs would be [2]

## References
1. Graves, Alex. "Adaptive computation time for recurrent neural networks." arXiv preprint arXiv:1603.08983 (2016).
2. Ballas, Nicolas, et al. "Delving deeper into convolutional networks for learning video representations." arXiv preprint arXiv:1511.06432 (2015).

**Questions:**

Overall, I think the proposed method is novel and has potential for impact, but given the issues mentioned in the weaknesses section, some more work is necessary for publication. I will consider to increase the score significantly should the issues be addressed during the author response period.

I have some minor comments regarding the write-up:
Line 63: starts upper-case after "and" in the previous item
Line 100 and 105: Shouldn't it be "4)" and "5)" instead of "2)" and "3)"? Either way, it is a bit confusing.
Equation 2: I'd explicitly state that $\mathbf{r}$ is the recurrent block and how $h_{-1}$ is defined. Also, I think it'd be good to explicitly state what $t$ and $t_\text{train}$ correspond to.
Line 197: "in Supp. Info" should maybe be "in the Appendix" or "in the supplementary material"
Line 266: "While ... whereas" ("while" or "whereas" should be removed)
Line 340-343: I'd avoid nesting parentheses.

## Acknowledgement of rebuttals
I have read the rebuttal and provided follow-up questions. The rebuttal addressed my concerns, except the addition of a strong non-recurrent baseline. I have accordingly increased the score slightly.

**Limitations:**

The authors state some limitations of the approach and propose directions for future research.

---

> ### Author Rebuttal · Authors · 2023-08-10
>
> 1. Our work is for the most part similar to the original ACT work, however, a key difference between our work and ACT is that our visual reasoning task involves static inputs whereas Graves (2016) can deal with variable-length sequences. Owing to this difference, our halting mechanism is the same as ACT applied to a 1-token input sequence, we will make this more explicit.
> 2. Line 188 is a typo in our initial submission, we did perform evaluation only on 3 implementations of recurrent computations (R-ResNet-30, Convolutional GRU, and LocRNN). Based on the newly added baselines we will update this statement to reflect the 5 total RNN cells used (including Linsley et al. (2018) and Linsley et al. (2020)).
> 3. All architectures were matched in terms of the number of parameters on PathFinder and Mazes evaluations respectively. For PathFinder, fix the number of kernels (d) to be 32 for LocRNN, 21 for ConvGRU and 64 for ResNet-30 with a filter size of 9x9 in the intermediate layers. For Mazes, we fix the number of kernels (d) to be 128 for LocRNN, ConvGRU, ResNet-30 and R-ResNet-30 and the kernel size is fixed to be 5x5.
> 4. In Table 2, R-ResNet-30 was equipped with ACT and it is a typo that we missed to mention ACT here. We have fixed this typo.
> 5. It would be very interesting to evaluate how well Vision Transformers perform on the two tasks we evaluate in our submission. Prior work (Tay et al. (2020)) showed that Transformers do not perform well on PathFinder. Given this and computational limitations, we were not able to test Vision Transformers on our versions of these tasks for the rebuttal.

---

> > ### Comment · Reviewer_1Kd6 · 2023-08-10
> > **Some items addressed, but**
> >
> > I have read all reviews, author responses, and considered the rebuttal pdf. I acknowledge that the authors addressed some of the issues mentioned in the weaknesses section of my review. However, since the authors only reported results of single runs (see item 9 in my review), it is difficult to judge how significant the differences between models are. Also, as mentioned in item 8, I think the consideration of additional strong non-recurrent baselines would've been important (ViTs were just an example). I will increase the score slightly to a borderline accept.

---

> > > ### Author Response · Authors · 2023-08-21
> > > **Results presented over multiple random initializations per reviewer request**
> > >
> > > Dear Reviewer,
> > >
> > > We have reported the performance of both LocRNN and ConvGRU (which are the two main models of comparison in our work) across 3 random initializations. We thank you for raising this issue, we find that (1) **LocRNN is consistently better performing than ConvGRU** and (2) LocRNN’s performance across random seeds is **more reliable with lesser variance when compared to ConvGRU**.
> > >
> > > We request you to please find the updated results in our common response posted on August 20 titled "LocRNN contrasted w/ (h)ConvGRU - more discussion and additional random seeds".
> > >
> > > We also add the part of our comment which discusses results across multiple random initializations here for your convenience.
> > >
> > > | **Model**       | **PathFinder-21** | **PathFinder-24** | **Mazes-19**     | **Mazes-25**     |
> > > |-----------------|-------------------|-------------------|------------------|------------------|
> > > | LocRNN (ACT)    | **92.89 +- 0.90**     | **85.81 +- 5.57**    |**86.83 +- 2.94**    | **49.99 +- 4.48**    |
> > > | ConvGRU (ACT)   | 82.63 +- 4.84     | 74.14 +- 6.52     | 75.1 +- 11.96    | 46.93 +- 4.2     |
> > >
> > > We will add these new results to our updated manuscript.
> > >
> > > NOTE:
> > > - All numbers above represent mean +- 1 SEM
> > > - Chance performance for PathFinder is 50%, whereas it is close to 0% for Mazes (where our performance metric measures the % of test-set mazes perfectly segmented)
> > >
> > > Thank you.

---

### Official Review · Reviewer_ikdz · 2023-07-12

**Soundness:** 3 good
**Presentation:** 3 good
**Contribution:** 2 fair
**Rating:** 6
**Confidence:** 4

**Summary:**

The primary subject of this paper is fostering computational efficiency in RNNs solving vision tasks by flexibly adapting the number of computing steps depending on the difficulty of the input. This is achieved through the addition of ACT (Adaptive Computation Time introduced by Graves, 2016) to recurrent vision models. The paper evaluates 4 models (3 with recurrence) on 2 visual reasoning tasks popular in the literature for studying RNN behavior (PathFinder and Mazes). One model is newly introduced here and given the name LocRNN. The key result put forward in the paper is that ACT-trained RNNs will indeed adaptively carry out more time steps before halting computation for difficult inputs (longer paths or larger mazes) than for easy inputs. Finally, another finding is that ACT-trained RNNs generalize to more difficult task conditions than those encountered during training.

**Strengths:**

The paper touches upon a significant and topical challenge in deep learning: how to avoid ‘wasting’ computations on easy inputs while still solving the hard ones to achieve computational efficiency. Efforts in that direction are valuable from an ecological perspective, among other perspectives.

Although the work doesn’t address this directly, mapping out which inputs are hard versus easy for a model can help acquire insights into the strategies a model has learned to solve a (visual) task. It is also of relevance for cognitive computational neuroscience and human-modal comparisons.

The paper reads very smoothly. The questions are formulated clearly, and so are the methods. It is overall well written.

**Weaknesses:**

It is good to see the prediction confirmed. Still, the finding that ACT will flexibly halt computation early for easier inputs does not tremendously advance the field, considering it’s what Graves (2016) already reported. With an ACT-like method (“DACT”) Eyzaguirre and Soto (2020) have already found this in visual reasoning tasks too. It’s worth a citation.

Similarly, there have been other thorough demonstrations that vision RNNs can generalize to more difficult task conditions, even on the very same Pathfinder task. I encourage the authors to check out and cite Linsley et al. (2020) "Stable and expressive recurrent vision models". In particular, Fig. 3 in this current paper is explored in detail and analyzed in the Linsley (2020) article.

The introduction of LocRNN then becomes the most novel aspect of the work, but there is not a lot of in-depth discussion in the main text about how it differs from prior models informed by neuroscience. At the very least, one would require a thorough comparison between the LocRNN and the model presented in Linsley et al. (2018) regarding their performance metrics and architectural similarities.

Finally, can the authors comment on whether this approach would work in other scenarios of visual difficulty apart from simply spatial extent? The notion of visual complexity involves a host of other factors which a non-hierarchical model would fail to account for, and which needs discussion.

Typos and minor fixes:

Some of the citations appear to be in the wrong format (e.g., L40)

L. 322: Fig.4 → Fig. 3?

Fig. 3: in a revised version, probably good to discuss these results in the main text more, specifically with regards to the non-ACT controls.

L 328: Fig. 4 has no panel c.

**Questions:**

Please refer to the weaknesses section above.

**Limitations:**

Please refer to the weaknesses section above.

---

> ### Author Rebuttal · Authors · 2023-08-10
>
> "It is good to see the prediction confirmed. Still, the finding that ACT will flexibly halt computation early for easier inputs does not tremendously advance the field, considering it’s what Graves (2016) already reported. With an ACT-like method (“DACT”) Eyzaguirre and Soto (2020) have already found this in visual reasoning tasks too. It’s worth a citation."
>
> 1. While we agree with the reviewer that Graves (2016) has shown that ACT will flexibly halt computation early for easier inputs, we would like to highlight that our unique contribution here is showing that it can be combined effectively with convolutional RNNs for improving performance on visual reasoning tasks. We thank the author for the relevant work by Eyzaguirre and Soto (2020). We shall add a citation to this work, but a key difference between this work and our submission here is that our models and evaluation are centered around extrapolation to novel difficulties on challenging visual reasoning problems. While we refer to [Kim et al 2018, Not-so-CLEVR] in support of the task used in Eyzaguirre and Soto (2020) as one known to not be challenging, it is certain from their experiments that they do not explore the scenario of generalizing to novel harder difficulty levels in their paper.
>
> "The introduction of LocRNN then becomes the most novel aspect of the work, but there is not a lot of in-depth discussion in the main text about how it differs from prior models informed by neuroscience. At the very least, one would require a thorough comparison between the LocRNN and the model presented in Linsley et al. (2018) regarding their performance metrics and architectural similarities."
>
> 2. Regarding comparison to Linsley et al. (2018) and Linsley et al. (2020), please refer to our comment addressing all reviewers where we discuss the difference between these works and our proposed work, as well as demonstrate our (empirical) high performance compared to models in both the above works.
>
> "Finally, can the authors comment on whether this approach would work in other scenarios of visual difficulty apart from simply spatial extent? The notion of visual complexity involves a host of other factors which a non-hierarchical model would fail to account for, and which needs discussion."
>
> 3. We also find it very interesting to test whether our proposed approach would scale to other scenarios of visual difficulty apart from spatial extent. Our work in progress and future scope will pursue this direction by studying AdRNNs on computer vision benchmarks based on natural images.
> 4. Thank you for highlighting typos in the paper, we shall promptly fix these issues in the submission.

---

> > ### Author Response · Authors · 2023-08-20
> > **Author discussion period is closing soon**
> >
> > Dear Reviewer,
> >
> > We would like to check if you have any concerns following our rebuttal and follow-ups that we may address during the author discussion period which ends on Aug 21st at 1 pm EDT.
> >
> > Thank you.

---

> > > ### Comment · Reviewer_ikdz · 2023-08-21
> > > **Thanks for your response!**
> > >
> > > Thanks for your response. It's interesting that the authors find that the LocRNNs are empirically superior (perhaps a more thorough suite of numerical evaluations including hyperparameter sweeps are necessary to ascertain this). I also do find the general discourse about stability as a halting criterion vs. learning explicit stop signals important and a relevant inclusion in the literature. I'm happy to update my evaluation. Good luck to the authors!

---

### Official Review · Reviewer_T9g1 · 2023-07-27

**Soundness:** 4 excellent
**Presentation:** 4 excellent
**Contribution:** 4 excellent
**Rating:** 6
**Confidence:** 3

**Summary:**

Authors show that using recurrent networks for adaptively processing static inputs for a variable number of iterations allows for zero-shot generalization to more difficult problems by simply unrolling the model predictions for more time steps at inference. Authors propose LocRNN and show model performance on the pathfinder and maze datasets, highlighting that their model uses a greater number of iterations for harder problems and can generalize to harder problems than those seen during training.

**Strengths:**

Clarity of Writing. This was by far my favorite paper to read of my reviewing batch because of the clarity of writing. I was able to clearly understand the premise and motivation and follow along with the experiment setup and evaluation.  The insight that recurrent networks can be used to scale up or down the amount of compute depending on the difficulty of the problem is well articulated.
Compelling Results. Authors show that LocRNN beats competitive baselines on two standardized benchmarks. Impressively, Table 2 shows that LocRNN beats other baselines on extrapolation datasets by a wide margin.


**Weaknesses:**

Motivation. Although I agree with the premise of the paper, I want to push back on using the biological inspiration of variable compute for neural networks. Although humans naturally can scale to harder and easier problems, state-of-the-art vision networks do not do this. Although authors cite efficiency as a reason for favoring RNNs for variable compute problems, no experimental evidence is provided.

Limited Baselines. Similarly, although the focus of this paper is on adaptive compute, two strong baselines would be LocRNN trained with only one computational step and LocRNN trained with N computational steps (where N is some fixed number of maximum steps allowed). Given the significant engineering that went into the network architecture of LocRNN, the ResNet-30 baseline does not seem convincing to show the necessity of adaptive compute.

Datasets. Although authors evaluate on two standard benchmarks, I would argue that these toy datasets do not represent real tasks. It would be more convincing to compete LocRNN against a state-of-the-art model on a more traditional computer vision benchmark (e.g. ImageNet, COCO, etc.) One place I think this approach could shine is in object detection. Given an image with a single near-field object, we should not need to spend a lot of compute. Similarly, given an image with many far-field small objects, we should likely spend more compute.


**Questions:**

Intermediate Predictions. How do we know that the iteration that the model stops at is optimal? Can we evaluate the model prediction at intermediate steps and evaluate the model prediction beyond when it naturally stops?
Architecture Diagram (Sec. 4.3). Providing an architecture diagram for LocRNN would help improve the clarity of Section 4.3. Please add this to the appendix.

**Limitations:**

Authors highlight that their work explore this problem of adaptive compute on static images and will work on videos in future work. However, I would encourage authors to explore more complex problems within static image domains (e.g. classification, detection, segmentation of natural images instead of toy-like environments).

---

> ### Author Rebuttal · Authors · 2023-08-10
>
> 1. Our experiments show that RNNs which scale computation are the only networks which are able to generalize to more challenging test samples by using more recurrent iterations at inference. We hypothesize that the vast levels of compute employed by SOTA networks today is thus driven by the long tail of the most difficult examples encountered during training and may be entirely unnecessary for the majority of the distribution. In a KDD23 plenary talk  (8/9/23) , Eric Horvitz  (CSO Microsoft) showed a plot of accuracy vs compute and showed that the last 1% accuracy of GPT-4 required 80% of the computational cost (through longer prompts).  Given the very real negative impact these massive computational systems have on the environment, there is strong motivation to reduce computation when possible. We have shown experimentally that with LocRNN and convGRU we can both reduce computation and increase accuracy because we’re not constrained to a one-size-fits-all level of compute like SOTA architectures.
>
> 2. While we agree with the reviewer that the end-goal is to inspect the performance of these systems in a naturalistic task, we intentionally conducted this study using tasks in which difficulty could be experimentally manipulated to study the generalization of different architectures in complexities beyond training. This is a lot more difficult in a dataset like ImageNet, in which organizing the examples by complexity is a non-trivial endeavor. We intend to pursue this direction in future work, but believe the chosen task represents an important middle ground between the simplistic tasks in prior literature and the end-goal of naturalistic images.

---

> > ### Comment · Reviewer_T9g1 · 2023-08-11
> >
> > Thanks for addressing my concerns in the rebuttal (adding other non-recurrent baselines and visualization of the LocRNN architecture) and adding additional context for your results. The exposition presented above strengthens the motivation for your experiments. I think this is a well written paper that should be recommended for acceptance.

---

### Author Rebuttal · Authors · 2023-08-10

Thank you all for taking the time to carefully review our paper. Here, we address common points based on overlapping comments in the reviews. We provide our responses to the constructive comments common to all reviewers in the following sections of our response:

### Differences with respect to Linsley et. al, 2020, Linsley et. al, 2018
CC: ikdz, cX6r
We thank the reviewers for bringing up this very relevant related work that we have by mistake not cited. We will certainly add a citation to Linsley et al, 2020, we agree that it is indeed relevant to our submission. We went one step further in including this model (hConvGRU trained with C-RBP) as well as the one in (Linsley et al, 2018) as part of our baselines using code made available by the authors (please find the updated results in Table 3 in the rebuttal). We find strong evidence in support of our model, despite tuning the above-mentioned models considerably, it seems very difficult to make them converge to a stable solution that better generalizes out of distribution on the tasks we used (PathFinder and Mazes) in comparison to LocRNN.

Differences between our papers:
1) @Reviewers ikdz, cX6r: It is important to note that  Linsley et al, 2020 uses a simplified version of PathFinder (the stimuli used look similar but the tasks have different rules) – The 2020 task involves densely supervised segmentation of the path with a circle on one of its ends whereas our task involves binary supervision of whether two circles are on the same path (which is the original Linsley et al 2018 task). The main difference is the information content of the supervision - the 2020 task provides pixel-by-pixel labeling, which is very informative and simple to learn as it enables learning to trace paths pixel-by-pixel; our task (following their prior work from 2018) provides only binary (sparse) classification information which is harder to learn.
Hence, the OOD generalization shown by Linsley et al, 2020 and what we show here (and used in Linsley et al, 2018) are different and not numerically comparable.

2) Linsley et al., 2018 & Linsley et al., 2020 use the hConvGRU RNN (in 2018 they train it with BPTT whereas in 2020, they train it with C-RBP). We add both these models to our evaluation baselines and in our experiments they don’t match the performance of LocRNN (see last three rows of the updated Table 3 in the Rebuttal). Despite increasing the number of recurrent iterations and using network output stability as the stopping criterion (as reviewer cX6r suggested), we find the network to not work well on Mazes 19x19 or Mazes 25x25 (both ConvGRU and LocRNN fare reasonably well on both these challenges). Considering that the same hConvGRU architecture learns these tasks while using BPTT supports the hypothesis that (a) hConvGRU+C-RBP is quite complex a model with many free parameters and hyperparameters making the network difficult to train (and also interpret), or (b) it is sensitive to particular random initializations where it works well.
3) Overall, with the above experiments we find clear and striking evidence of the novelty and improved performance of LocRNN with respect to hConvGRU (Linsley et al, 2018), and hConvGRU + C-RBP (Linsley et al, 2020).

### Using stability as a halting criterion as opposed to learning halting
We considered using network stability as a criterion for halting.
We found this to work suboptimally for a number of reasons.
1) Avoiding hand-engineered heuristics: Most importantly, it requires defining a hand-engineered heuristic on how much change in the output is considered small enough to halt. In typical segmentation tasks, the network output for the initial few timesteps is highly stable in making nonsensical predictions (predicting all pixels as the negative class) and thus, heuristics need to identify an inflection point in the output trajectory where meaningful predictions start to emerge and stabilize. In the absence of ground truth information, one cannot pick a heuristic that generalizes to unseen data.
2) Avoiding hard assumptions / constraints on stable hidden states: Learnable halting makes less assumptions about the hidden state’s properties, and hence doesn’t enforce hard constraints such as stability to be satisfied by training. Some, but not all RNNs, have stable hidden states wherein the network response stops changing after reaching an attractor. In fact, Linsley et al., 2020 that reviewer cX6r refers to argue that RNNs that are expressive have an intrinsic inability to learn stable hidden states.

For RNNs to be stable, their hidden state transformation needs to model a contractive mapping (Miller & Hardt, 2019; Pascanu et al, 2013). That is, the recurrent transition function F satisfies the following inequality:
			||F(h_t) - F(h_{t-1})||_p < \lambda ||h_t - h_t-1||_p
RNNs with stable hidden states that satisfy the above inequality are quite difficult to train on challenging problems (these kinds of networks are similar in nature to Deep Equilibrium Models [1] and Neural ODEs [2] which are not as successful yet as RNNs in their ability to solve a wide variety of tasks) as there is an inherent tradeoff between stability of the hidden state and expressivity (Linsley et al., 2020). Even when stable models perform comparably to unstable models as in Miller & Hardt, 2019, the authors show unstable models’ advantages such as performance improvements in the short-time horizon and lesser vanishing gradient issues.
3) Finally models operating on dynamic input (as in the real world) may not be expected (or desired) to reach a stable hidden state.
4) We also tried the  Linsley et al 2020 algorithm with a threshold on state change and it performed very poorly on Mazes (see Table 3 in the Rebuttal).

---

### Author Response · Authors · 2023-08-10
**More about LocRNN architecture**

### LocRNN architecture and why it works well
With respect to  LocRNN’s functioning, and its differences to other similar architectures such as ConvGRU, and hConvGRU (Linsley et al., 2018; Linsley et al., 2020), we have (1) Run experiments comparing these different architectures, (2) Described the origin of the dynamics of LocRNN in further detail, and (3) Qualitatively visualized the internals of LocRNN.

We would like to highlight that the dynamics of LocRNN originate from a previously published model of cortical horizontal connections by Zhaoping Li (Li, 1998). The PDEs proposed in (Li, 1998) model (in continuous-time) the connections between V1 neurons. We take these PDEs, apply Euler integration to them in order to convert them into discrete-time difference equations which we then implement using convolutional Recurrent Neural Networks. The final RNN obtained from Li 1998 after Euler integration is what we refer to as LocRNN in our submission, and we contrast the architectures between LocRNN and GRU in Fig 8.

---

### Author Response · Authors · 2023-08-20
**LocRNN contrasted w/ (h)ConvGRU - more discussion and additional random seeds**

Dear Reviewers,

We thank you for your quick responses to our first rebuttal. Following up on Reviewer 1Kd6’s request, we report the performance of both LocRNN and ConvGRU (which are the two main models of comparison in our work) across 3 random initializations. We thank the reviewer for raising this issue, we find that (1) LocRNN is consistently better performing than ConvGRU and (2) LocRNN’s performance across random seeds is more reliable with lesser variance when compared to ConvGRU.

### Why LocRNN works better than baselines
Our initial exploration of this direction so far has suggested the following for LocRNN’s improved performance and the importance of its various components:

We have found that the most important component of LocRNN is the incorporation of “hidden” (S)  interneurons (not directly connected to the output readout) absent in simpler RNNs like GRU. Theoretically, we are motivated in this interpretation by the importance of hidden neurons in deep neural networks and the difference in performance between Jordan (output feedback) and Elman nets (hidden state feedback).   Empirically, removing this feature from LocRNN results in reduced performance, which we hypothesize is why LocRNN performs better than ConvGRU.    On a more specific note, Fig 7 in the supplementary rebuttal pdf shows state activations of the L (projection) population for a PathFinder example showing how the L neurons perform contour integration; we have similarly inspected the S (interneuron) population which have clearly different activations but are not as easy to interpret. These interneuron-analogues appear to be encoding complementary information that we are continuing to investigate.

\textit{To explain the difference between LocRNN and the more complex hConvGRU we believe the following are critical.}
LocRNN explicitly uses two separate populations of **laterally-connected** neurons inspired by cortical evidence of different horizontal connection patterns in V1 between interneuron and projection neurons.
hConvGRU’s hidden state is split into H1[t] and H2[t], however, they are **hierarchically organized** unlike LocRNN (which is derived from Zhaoping Li’s model of contour integration in V1 where the two populations are laterally-connected).
Extra connectivity and gating connections beyond those required to have separate interneuron and projection neurons appear to result in poorer performance due to the increased (potentially unnecessary) complexity
In summary, LocRNN uses a unique implementation of lateral connections that is more faithful to biological projection and inter- neurons.

In summary, our working hypothesis is that our network forms something like a “smallest necessary core” for good performance in these extrapolation tasks. As this is a critical question for RNNs in general, we will be further exploring these ideas in future work.

| **Model**       | **PathFinder-21** | **PathFinder-24** | **Mazes-19**     | **Mazes-25**     |
|-----------------|-------------------|-------------------|------------------|------------------|
| LocRNN (ACT)    | **92.89 +- 0.90**     | **85.81 +- 5.57**    |**86.83 +- 2.94**    | **49.99 +- 4.48**    |
| ConvGRU (ACT)   | 82.63 +- 4.84     | 74.14 +- 6.52     | 75.1 +- 11.96    | 46.93 +- 4.2     |

We will add these new results to our updated manuscript.

NOTE:
- All numbers above represent mean +- 1 SEM
- Chance performance for PathFinder is 50%, whereas it is close to 0% for Mazes (where our performance metric measures the % of test-set mazes perfectly segmented)

---

### Decision · Program_Chairs · 2023-09-21

**Decision:**

Accept (poster)

**Comment:**

This paper explores the problem of generalization to unseen visual reasoning  tasks, through learning to allocate computation time that is proportional to the task's difficulty. The authors also proposed a novel architecture, LocRNN.
The reviewers initially had concerns about the impact of the architecture and the statistical significance of the results. The authors provided in-depth complimentary experiments which shed more light on the mechanism, thus making the submission stronger.